# AN INVESTIGATION OF BATCH NORMALIZATION IN OFF-POLICY ACTOR-CRITIC ALGORITHMS

## ABSTRACT

Batch Normalization (BN) has played a pivotal role in the success of deep learning by improving training stability, mitigating overfitting, and enabling more effective optimization. However, its adoption in deep reinforcement learning (DRL) has been limited due to the inherent non-i.i.d. nature of data and the dynamically shifting distributions induced by the agent's learning process. In this paper, we argue that, despite these challenges, BN retains unique advantages in DRL settings, particularly through its stochasticity and its ability to ease training. When applied appropriately, BN can adapt to evolving data distributions and enhance both convergence speed and final performance. To this end, we conduct a comprehensive empirical study on the use of BN in off-policy actor-critic algorithms, systematically analyzing how different training and evaluation modes impact performance. We further identify failure modes that lead to instability or divergence, analyze their underlying causes, and propose the Mode-Aware Batch Normalization (MA-BN) method with practical actionable recommendations for robust BN integration in DRL pipelines. We also empirically validate that, in RL settings, MA-BN accelerates and stabilizes training, broadens the effective learning rate range, enhances exploration, and reduces overall optimization difficulty. Our code is available at: `https://anonymous.4open.science/r/bn_rl468305485`.

## 1 INTRODUCTION

Batch Normalization (BN) (Ioffe & Szegedy, 2015) has been a foundational technique in deep learning, playing a critical role in improving training stability(Santurkar et al., 2018), introducing stochasticity (Shekhovtsov & Flach, 2018; Huang et al., 2020), and enabling domain adaptation(Wang et al., 2020; Schneider et al., 2020). It has become a milestone in the development of deep neural networks due to its effectiveness in mitigating overfitting. However, its adoption in deep reinforcement learning (DRL) has remained limited. Unlike supervised learning, where data is typically assumed to be independent and identically distributed (i.i.d.) and drawn from a fixed distribution, online DRL introduces unique challenges: the data is inherently non-stationary and temporally correlated, as it is generated online by agents whose learning continuously alters the data distribution. Violations of the i.i.d. assumption complicate the use of BN, which depends on stable batch statistics. These challenges have limited BN's adoption in reinforcement learning (RL). Early RL benchmarks were relatively simple, with shallow architectures that could be trained effectively even without BN. As tasks and networks grow in complexity, stable optimization and effective regularization become crucial. This raises the question of whether BN can meaningfully support training in modern RL.

While Layer Normalization (LN) (Ba et al., 2016) has been increasingly adopted in the RL community (Makoviichuk & Makoviychuk, 2021; Lyle et al., 2024; Zhong et al., 2024), its normalization operates independently of the input data distribution, making it more robust to the distributional shifts commonly encountered in RL. We argue that, when properly configured, BN can be not only a viable but also a beneficial choice in DRL. BN smooths the loss landscape and improves the optimization conditions, reducing training difficulty. Its inherent stochasticity enhances exploration, improves generalization, and mitigates overfitting, which is particularly important in RL, especially for offline RL and sim-to-real transfer. Empirical evidence further highlights this potential: CrossQ (Bhatt et al., 2024) achieves state-of-the-art performance by applying Batch Renormaliza-

tion (BRN) (Ioffe, 2017)[1] in both the actor and critic networks while removing the target network, outperforming LN-based variants. Taken together, these findings indicate that BN and its variants, despite their challenges, can provide distinctive advantages in modern RL if applied with care. A critical but underexplored aspect is the selection of training versus evaluation modes for BN, which behave differently due to their reliance on batch versus running statistics. Prior works such as Deep Deterministic Policy Gradient (DDPG) (Lillicrap et al., 2015), CrossNorm (Bhatt et al., 2019), and CrossQ have employed BN or its variants without systematically investigating this choice, leaving a gap in our understanding of its practical implications. We empirically study BN mode selection in off-policy actor-critic methods, uncover divergence-inducing configurations, and provide practical guidance for stable BN usage in DRL. Our contributions are threefold:

- **Systematic investigation:** We systematically investigate the role of BN training and evaluation modes in off-policy actor-critic frameworks.
- **Analysis of divergence:** We identify configurations that cause instability and analyze the underlying reasons for divergence.
- **Practical guidelines and advantages:** We propose the Mode-Aware Batch Normalization (MA-BN) method with practical mode selection guidelines, highlighting its benefits in RL: accelerated and stabilized convergence, broader learning rate range, enhanced exploration, and reduced optimization difficulty.

## 2 BACKGROUND

### 2.1 DEEP REINFORCEMENT LEARNING

Reinforcement learning problems are typically formulated as Markov decision processes (MDPs), defined by the tuple $(\mathcal{S}, \mathcal{A}, P, r, \gamma, \rho_0)$. Here, $\mathcal{S}$ and $\mathcal{A}$ denote the state and action spaces, and $\Delta(\mathcal{X})$ denotes the probability simplex over $\mathcal{X}$; $P : \mathcal{S} \times \mathcal{A} \to \Delta(\mathcal{S})$ is the transition distribution; $r : \mathcal{S} \times \mathcal{A} \to \mathbb{R}$ is the reward function; $\gamma \in [0, 1)$ is the discount factor; and $\rho_0 \in \Delta(\mathcal{S})$ is the initial state distribution. The agent aims to learn a policy $\pi : \mathcal{S} \to \Delta(\mathcal{A})$ that maximizes the expected cumulative discounted return: $\mathbb{E}_\pi[\sum_{t=0}^{\infty} \gamma^t r(s_t, a_t)]$.

### 2.2 OFF-POLICY ACTOR-CRITIC ALGORITHMS

Actor-critic algorithms jointly learn a parameterized policy $\pi$ (the *actor*) and an action-value function $Q$ (the *critic*), where the critic estimates expected returns for state–action pairs and the actor is optimized to maximize the critic's evaluation. Off-policy algorithms decouple data collection from policy updates by leveraging experience replay, which greatly improves sample efficiency. A canonical example is DDPG, which combines Deterministic Policy Gradient (Silver et al., 2014) with Q-learning and target networks. The critic is trained by minimizing the Bellman error:

$$\min_\theta J_\theta(\mathcal{D}) = \mathbb{E}_{(s_t,a_t,r_t,s_{t+1})\sim\mathcal{D}} \left[ (Q_\theta(s_t, a_t) - r_t - \gamma Q_{\bar{\theta}}(s_{t+1}, \pi_\phi(s_{t+1})))^2 \right], \quad (1)$$

and the actor is optimized by solving:

$$\phi^\star = \arg\max_\phi \mathbb{E}_{s_t\sim\mathcal{D}} \left[ Q_\theta(s_t, \pi_\phi(s_t)) \right]. \quad (2)$$

SAC (Haarnoja et al., 2018) improves upon DDPG by introducing entropy regularization, which promotes stochastic exploration and enhances stability. SAC-Discrete (SACD) (Christodoulou, 2019) extends SAC to discrete action spaces, but often suffers from instability and Q-value over-estimation. Stable Discrete SAC (SD-SAC) (Zhou et al., 2022) addresses these issues by combining entropy penalties with double Q-learning, target smoothing, and Q-value clipping, substantially improving stability in discrete domains. Building on SAC, DRQ (Yarats et al., 2021b) incorporates a convolutional encoder and random-shift augmentation, while DRQv2 (Yarats et al., 2021a) further improves performance by replacing SAC with DDPG as the base algorithm and introducing an exploration schedule, as shown in Equation 3:

$$\sigma(t) = \sigma_{\text{init}} + (1 - \min(\tfrac{t}{T}, 1))(\sigma_{\text{final}} - \sigma_{\text{init}}) \quad (3)$$

---

[1]BRN is a variant of BN that blends batch statistics with moving averages (population statistics) to improve training stability under small batches or non-stationary settings.

---

**Algorithm 1** Off-policy Actor-Critic Update Routine

---

**Require:** Encoder $f_\xi$, Policy $\pi_\phi$, Q-functions $Q_{\theta_1}, Q_{\theta_2}$
**Require:** Experience replay buffer $\mathcal{D}$
1: Sample a batch $\tau = (s_t, a_t, r_t, s_{t+1}) \sim \mathcal{D}$
2: **Update Critic:**
3:    $a_{t+1} \leftarrow \pi_\phi(s_{t+1}) + \epsilon, \epsilon \sim \text{clip}(\mathcal{N}(0, \sigma^2), -c, c)$              (Actor-II)
4:    $y \leftarrow r_t + \gamma \min_{k=1,2} Q_{\theta_k}(s_{t+1}, a_{t+1})$                 (Critic-III)
5:    $\mathcal{L}_{\theta_k, \xi} \leftarrow \mathbb{E}_\tau \left[ (Q_{\theta_k}(s_t, a_t) - y)^2 \right]$               (Critic-II)
6: **Update Actor:**
7:    $a'_t \leftarrow \pi_\phi(s_t) + \epsilon, \epsilon \sim \text{clip}(\mathcal{N}(0, \sigma^2), -c, c)$                (Actor-I)
8:    $\mathcal{L}_\phi \leftarrow -\mathbb{E}_{s_t} \left[ \min_{k=1,2} Q_{\theta_k}(s_t, a'_t) \right]$                (Critic-I)

---

Here, $\sigma_{\text{init}}$ and $\sigma_{\text{final}}$ denote the initial and final standard deviations, and $T$ is the decay horizon. With better hyperparameter choices, such as a larger replay buffer, DRQv2 achieves strong performance on DMC, surpassing other state-of-the-art agents. Notably, it is the first model-free method to solve challenging humanoid tasks directly from pixels. More recently, CrossQ (Bhatt et al., 2024) improves continuous control by removing target networks and employing Batch Renormalization with wider critic networks. These design choices enhance sample efficiency and stability by implicitly regularizing the critic and simplifying the update pipeline. Overall, despite their design differences, most off-policy actor–critic algorithms are built upon the same underlying framework, and their basic update procedure can be summarized in Algorithm 1[2].

## 2.3 BATCH NORMALIZATION

Batch Normalization is a widely adopted technique in deep learning for stabilizing and accelerating training. Given an activation $x$ from a neuron in a deep network layer, BN normalizes it over a mini-batch of size $m$ during training as:

$$\hat{x}^{(i)} = BN_{\text{Train}} = \gamma \frac{x^{(i)} - \mu}{\sqrt{\sigma^2 + \epsilon}} + \beta, \quad i = 1, \dots, m, \qquad (4)$$

where $\mu = \frac{1}{m} \sum_{i=1}^m x^{(i)}$, $\sigma^2 = \frac{1}{m} \sum_{i=1}^m (x^{(i)} - \mu)^2$, $\epsilon$ is a small constant for numerical stability, and $\gamma, \beta \in \mathbb{R}$ are learnable scale and shift parameters. Since each mini-batch is sampled randomly at every training step, the statistics $\mu$ and $\sigma^2$ are themselves random variables (Huang et al., 2019; Teye et al., 2018), causing the normalized output $\hat{x}^{(i)}$ to vary depending on the batch composition. During inference, BN replaces these batch statistics with running estimates of the population statistics $\{\hat{\mu}, \hat{\sigma}^2\}$, updated throughout training via:

$$\begin{cases} \hat{\mu} \leftarrow (1 - \lambda)\hat{\mu} + \lambda\mu \\ \hat{\sigma}^2 \leftarrow (1 - \lambda)\hat{\sigma}^2 + \lambda\sigma^2 \end{cases} \qquad (5)$$

When evaluating, the normalization uses the evaluation mode:

$$\hat{x}^{(i)} = BN_{\text{Eval}} = \gamma \frac{x^{(i)} - \hat{\mu}}{\sqrt{\hat{\sigma}^2 + \epsilon}} + \beta. \qquad (6)$$

BN enhances training stability by smoothing gradients and enabling larger learning rates, while also acting as a stochastic regularizer that improves generalization. However, when applied to RL, where data are inherently correlated and distributions evolve over time, BN must be carefully adapted; otherwise, shifts in the estimated statistics (Huang et al., 2022) can destabilize training and lead to performance collapse.

---

[2]Entropy-regularized variants further augment the objective with an additional entropy term.

# 3 ANALYSIS OF BATCH NORMALIZATION MODE SELECTION IN OFF-POLICY ACTOR-CRITIC ALGORITHMS

Unlike in supervised learning, the distinction between training and evaluation modes for BN is less clear in RL, where various mode combinations may be reasonable. Prior works such as DDPG and CrossQ have applied BN with specific mode settings but without analyzing their rationale or implications. To guide proper BN usage in RL, we systematically study the impact of BN mode selection within the off-policy actor-critic framework (Algorithm 1), keeping all other hyperparameters fixed for fair comparison, and analyze the BN mode configuration at each step. For clarity, we denote T as the training mode (Eq.4), in which batch normalization is performed using batch statistics; E as the evaluation mode (Eq.6), in which batch normalization is performed using running statistics; and Origin to indicate the absence of normalization. The configuration for the critic is represented as a three-letter sequence specifying the modes of Critic-I, Critic-II, and Critic-III in order, whereas the configuration for the actor is expressed as a two-letter sequence specifying the modes of Actor-I and Actor-II. For instance, 'TET' indicates that Critic-I, Critic-II, and Critic-III operate in training, evaluation, and training modes, respectively, while 'TE' denotes that Actor-I and Actor-II operate in training and evaluation modes.

## 3.1 ANALYSIS OF MODE SELECTION FOR CRITIC-I

We investigate the impact of BN mode selection in step Critic-I by comparing three configurations: Critic-I T, Critic-I E, and Origin. We fix the mode selections of Critic-II and Critic-III to training mode (see Subsection 3.2 for analysis). For consistency, we use the configuration where BN is applied after the linear layer as a representative case in our analysis. As shown in Fig. 1, Critic-I E leads to stable convergence and outperforms the Origin baseline, whereas Critic-I T significantly degrades performance, emphasizing the importance of proper BN mode selection for effective policy learning. Additional experiments with different algorithms and BN placements (Appendix A) yielded consistent results, reinforcing that the observed discrepancy stems from the distinct behaviors of Critic-I T and Critic-I E. We also compare normalized Q prediction biases in Fig. 1, as discussed in (Hiraoka et al., 2021; Chen et al., 2021; Bhatt et al., 2024). As observed, the failure of Critic-I T is attributed to inaccurate Q-value estimation, with both the mean and variance of the Q-value bias being significantly higher than in the other two configurations.

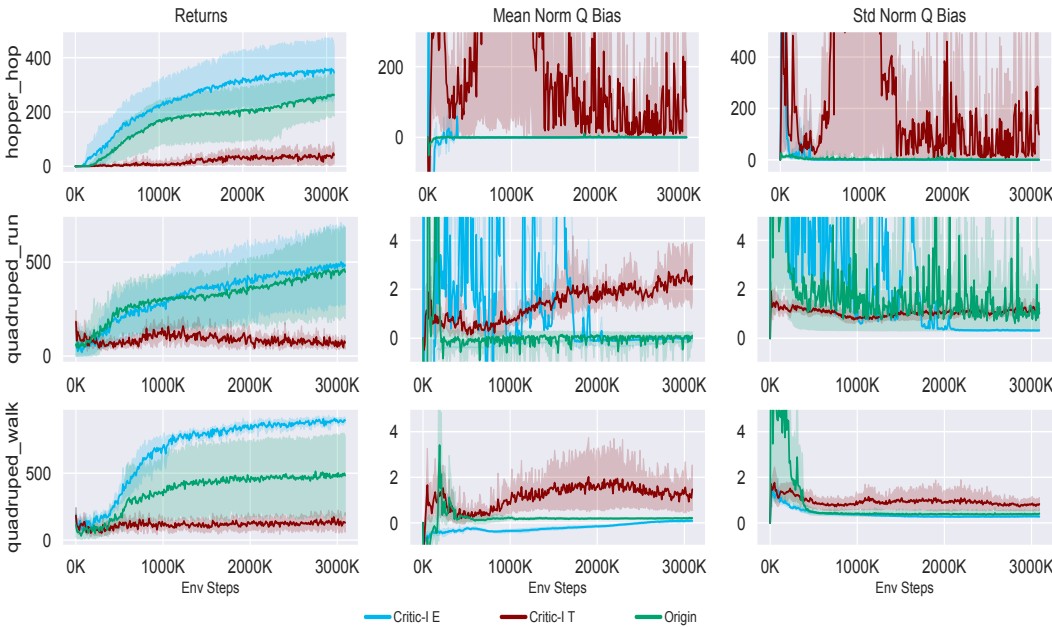

Figure 1: Return curves of the DRQv2 algorithm on DMC tasks under different settings.

A natural hypothesis is that, under BN's training mode, the statistical fluctuations arising from small batch sizes induce variability in Q-value estimates for the same input across different mini-batches. Such variability increases the variance of the resulting gradient estimates, which can destabilize policy optimization and ultimately degrade learning performance. For analytical convenience, we formalize this intuition in the following theorem regarding the action distribution in the replay buffer (see Appendix B for proof).

**Theorem 1.** *Assume the actor policy at time $t$ is Gaussian, $a \sim \mathcal{N}(\mu_t, \delta_t^2 I_d)$ with $\mu_t \in \mathbb{R}^d$ and $\delta_t^2 > 0$. At each time step, the actor generates $C$ actions, resulting in $C$ transitions $\{(s_t, a_t^{(c)}, r_t^{(c)}, s_{t+1}^{(c)})\}_{c=1}^C$, which are stored in the replay buffer. The buffer capacity is $N$ times larger than a single-step batch, i.e., it holds $NC$ transitions in total. Assume the policy evolves smoothly over time: $\|\mu_t - \mu_{t+1}\|_2 \leq \Delta_\mu$, $|\delta_t^2 - \delta_{t+1}^2| \leq \Delta_\sigma$. Then:*

1. *The pairwise differences of actor parameters are bounded as:*

$$\max_{i,j} \|\mu_i - \mu_j\|_2 \leq N\Delta, \quad \max_{i,j} |\delta_i^2 - \delta_j^2| \leq N\Delta.$$

2. *Let $a$ be a randomly sampled action from the buffer, and let $\bar{a}$ be the empirical mean of a batch of $B$ i.i.d. samples from the buffer. Then:*

$$\mathbb{E}[a] = \mathbb{E}[\bar{a}] = \mu_a, \quad Var(a) = \sigma_a^2, \quad Var(\bar{a}) = \sigma_a^2/B,$$

*where:*

$$\mu_a = \frac{1}{N} \sum_{t=1}^N \mu_t, \quad \sigma_a^2 = \frac{1}{N} \sum_{t=1}^N \delta_t^2 + \frac{1}{N} \sum_{t=1}^N \|\mu_t - \mu_a\|_2^2.$$

By Theorem 1, the variability of batch-level statistics can be characterized by the variance term $\sigma_a^2$, which depends on both the buffer size and the batch size. Specifically, if the fluctuation of batch-level statistics is the primary reason for the poor performance of Critic-I-T, then increasing the batch size or decreasing the buffer size should gradually improve its performance. We conduct experiments to test this hypothesis. As illustrated in Fig. 2, increasing the batch size and reducing the buffer size have little effect on the performance of Critic-I T. In contrast, Critic-I E consistently achieves good results under most settings. This suggests that the failure of Critic-I T to converge is not primarily caused by fluctuations in batch statistics, indicating the presence of additional underlying factors that warrant further investigation. In the `quadruped_run` environment, Critic-I E performs poorly with a small replay buffer, likely due to strong sample correlations destabilizing off-policy learning, highlighting the need for sufficiently large buffers to ensure stable training.

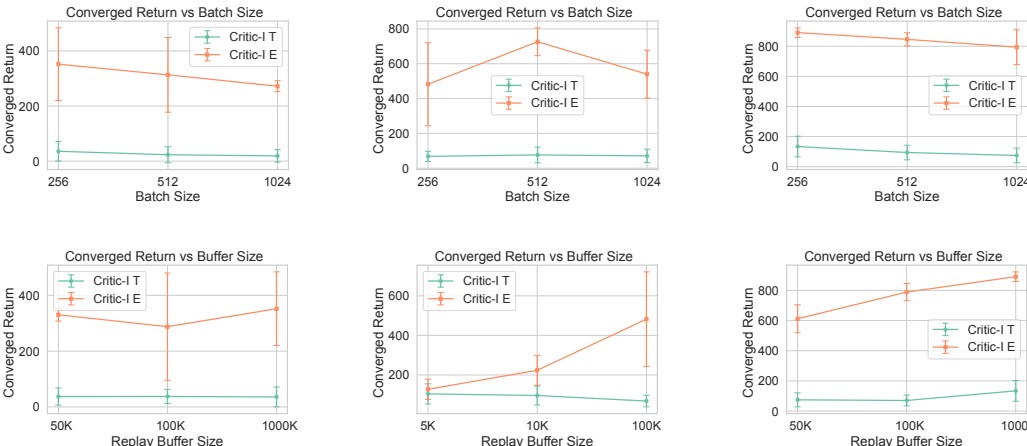

Figure 2: Comparison of converged returns in DMC tasks using the DRQv2 algorithm. Top row shows the effect of varying batch sizes, and bottom row shows the effect of varying buffer sizes. From left to right: hopper_hop, quadruped_run, quadruped_walk.

To identify the reason for the poor performance of Critic-I T, we further analyze the distributional differences in the data. Under our setting, algorithm 1 trains the critic in Step Critic-II with BN in training mode, sampling both $s_t$ and $a_t$ from the replay buffer. As a result, the critic becomes well-adapted to the data distribution stored in the buffer. However, in Step Critic-I, when computing the Q-values for the actor update, the action $a'_t$ is sampled from the current actor policy, which may significantly deviate from the distribution of actions stored in the replay buffer. According to Theorem 1, the discrepancy between the actor-generated action and those in the buffer can be bounded by $N\Delta$, where the bound scales with the buffer size. Consequently, when the buffer is large, this mismatch can become substantial. If BN remains in training mode during Step Critic-I, the normalization statistics are still governed by the buffer distribution, effectively masking the distributional shift of the updated actor. This mismatch can lead to inaccurate Q-value estimation and, consequently, suboptimal or even detrimental policy updates for the actor.

To empirically validate our hypothesis, we conducted a mitigation experiment for Critic-I T by introducing a data mixing strategy. Specifically, during Step Critic-I where the actor relies on the critic to estimate Q-values, we augmented the mini-batch with additional state-action pairs $(s_{bt}, a_{bt})$ sampled from the replay buffer, alongside the original samples $(s_t, a'_t)$ generated by the current actor policy. Unlike CrossNorm, we explicitly disable gradient computation for the auxiliary samples $\{(s_{bt}, a_{bt})\}$, using them exclusively to update BN statistics in training mode. We define the buffer mixing ratio as $1 : x$, where $x$ represents the number of actor-generated samples per buffer sample. We refer to this configuration as `buffer_x`. As the proportion of buffer data increases, the batch distribution in Step Critic-I gradually aligns with that of the replay buffer, thereby stabilizing the BN statistics. Consequently, we observe improved convergence behavior. As shown in Figure 3, higher buffer mixing ratios lead to faster convergence and higher final performance under training mode. Notably, when the ratio reaches $1 : 2$, the performance approaches or even surpasses that achieved in evaluation mode, reinforcing our claim that distribution mismatch is a key factor behind the degradation of training-mode performance. We also report quantitative analyses of distributional shifts and Q-value estimation bias under the mixed mode; detailed results are provided in Appendix C. We further examined the impact of Critic-I mode selection within BRN for CrossQ. Since CrossQ does not discuss mode selection, we analyzed its Critic-I modes and obtained conclusions consistent with both those reported above and CrossQ (see Appendix D). For discrete-action tasks, we evaluated the SD-SAC algorithm; due to the relatively small differences in the discrete action space, both Critic-I T and Critic-I E achieved comparable performance (see Appendix E). In addition, we observe that Critic-I T can amplify the actor's local policy biases, worsening estimation errors and potentially triggering a self-reinforcing failure loop (see Appendix F for an illustrative example).

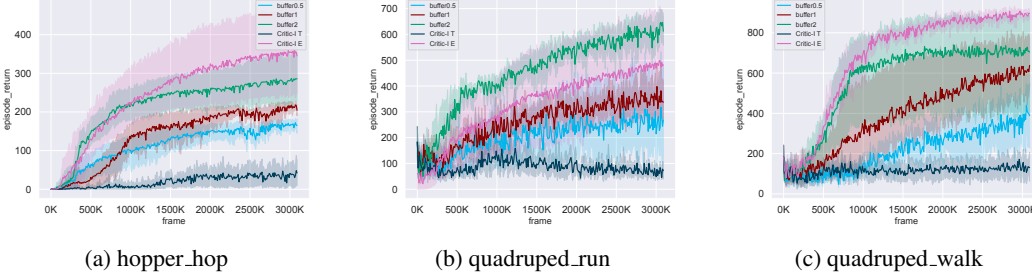

| (a) hopper_hop | (b) quadruped_run | (c) quadruped_walk |

Figure 3: Performance comparison of DRQv2 under different buffer mixing ratios in training mode.

## 3.2 ANALYSIS OF MODE SELECTION FOR CRITIC-III

For Step Critic-II, the critic is updated using target values from the target critic. Since this step is inherently part of the learning process and requires updating the BN statistics, Critic-II T is naturally employed and will not be discussed further. We therefore focus our analysis on Step Critic-III, which leverages the target critic to compute stable target Q-values. While evaluation mode can offer more stable Q estimates due to fixed BN statistics, the soft update mechanism used to maintain the target critic introduces a potential mismatch between its parameters and BN statistics, which may undermine the reliability of the target Q-values. For empirical validation, we use Critic-I E to provide accurate Q-value estimates for actor training, and disable BN in the actor. For Step Critic-I to Critic-

III, we then evaluate several configurations: (1) ETT, and (2) ETE, with three BN update strategies for the target critic—no update (ETE+BN-0), soft update along with parameters (ETE+BN-soft), direct copying from the critic (ETE+BN-critic). As shown in Figure 4, ETT achieves superior performance in two tasks and ETE+BN-soft/critic achieve better performance in one task. In contrast, ETE+BN-0 fails due to stale BN statistics. While ETE+BN-soft and ETE+BN-critic partially synchronize the BN statistics with the target network parameters, thereby improving performance to some extent, they do not completely eliminate the mismatch. Due to the complex nonlinear transformations in deep networks, parameter updates (via soft update) and BN statistic updates may evolve at different scales and dynamics, leading to residual misalignment between the statistics and the underlying parameter distribution. However, using BN in evaluation mode stabilizes the target critic by fixing its statistics, may reducing variance and enhancing training stability, which may work well in some cases. Thus, the choice between Critic-III T and Critic-III E depends on task-specific characteristics. For simplicity, we adopt the Critic-III T as default setting.

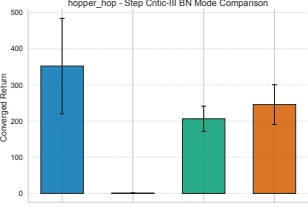 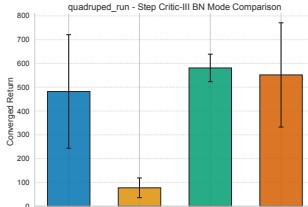 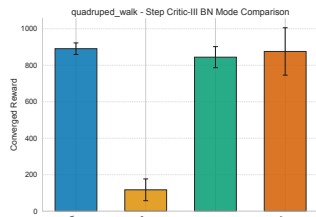

Figure 4: Performance comparison of DRQv2 under different mode setting in critic training. From left to right: hopper_hop, quadruped_run, quadruped_walk.

## 3.3 ANALYSIS OF MODE SELECTION FOR ACTOR

In most off-policy actor-critic algorithms, BN in the actor network participates in both Actor-I and Actor-II steps, influencing action selection and exploration during updates of both the actor and the critic. Besides, using either training or evaluation mode for BN in the actor is reasonable. To isolate the effect of BN in the actor, we disable BN in the critic to remove potential confounding factors. We then experiment with four actor network configurations: "TT", "TE", "ET", and "Origin". The "EE" configuration is excluded, as the absence of batch normalization statistics updates due to the lack of training mode usage degenerates the BN layer into a fixed affine transformation, providing no normalization effect. We evaluate

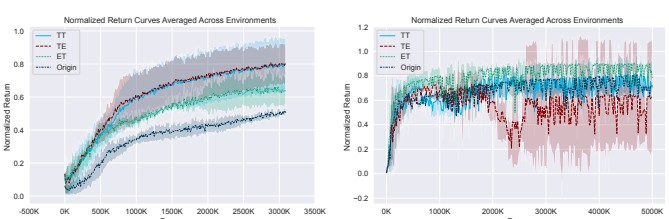

Figure 5: Actor network convergence under different BN configurations: (**Left**) DRQv2 on DMC (hopper_hop, quadruped_run, quadruped_walk); (**Right**) SAC on Mujoco (Humanoid-v4, Walker2d-v4).

DRQv2 and SAC on the DMC and Mujoco benchmarks, respectively, with training curves shown in Figure 5. Incorporating BN has the potential to achieve faster convergence and better final performance, though the optimal configuration varies across algorithms. In DMC, TT and TE achieve the best results; in Mujoco, ET performs best, while TT remains comparable to the baseline. Overall, the TT mode yields strong results in most cases while maximizing stochasticity, and we therefore adopt it as the default configuration. Since CrossQ does not discuss mode selection within BRN, we analyze its actor network and obtain similar conclusions (see Appendix G).

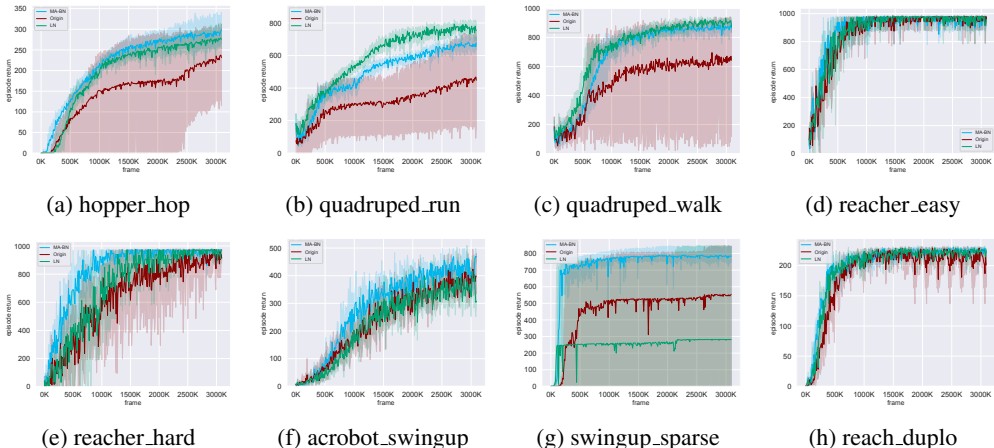

(a) hopper_hop    (b) quadruped_run    (c) quadruped_walk    (d) reacher_easy

(e) reacher_hard    (f) acrobot_swingup    (g) swingup_sparse    (h) reach_duplo

Figure 6: Return curves across 8 DMC environments under different regularization methods using DRQv2. Note: swingup_sparse = cartpole_swingup_sparse.

## 4 EVALUATING MA-BN IN OFF-POLICY ACTOR-CRITIC ALGORITHMS

Based on our previous analysis, we set the BN modes of the actor and critic networks to "TT" and "ETT", respectively, and refer to this configuration as Mode-Aware Batch Normalization (MA-BN). We then empirically evaluate MA-BN against alternative BN settings.

### 4.1 COMPARISON OF NORMALIZATION STRATEGIES IN DRQV2 ON DMC

We evaluate MA-BN against LN and no normalization using DRQv2 across eight DMC tasks. As shown in Figure 6, MA-BN consistently achieves superior performance in most environments, characterized by faster convergence and higher final returns. By introducing beneficial stochasticity, MA-BN also has the potential to accelerate exploration, thereby leading to improved outcomes. This observation is consistent with findings reported in CrossQ, further corroborating the potential of MA-BN to yield enhanced convergence behavior under appropriate configurations.

### 4.2 POTENTIAL INFLUENCE OF BATCH NORMALIZATION ON EXPLORATION

Due to the stochasticity introduced by the batch statistics in BN, it has the potential to accelerate exploration. To investigate this effect, we designed the maze environment illustrated in Figure 7 (left), where the agent must navigate from the start to the goal while avoiding obstacles and a designated death zone. The action space is continuous, and we employ the DDPG algorithm to train the agent under two conditions: Origin and MA-BN, keeping all other hyper-

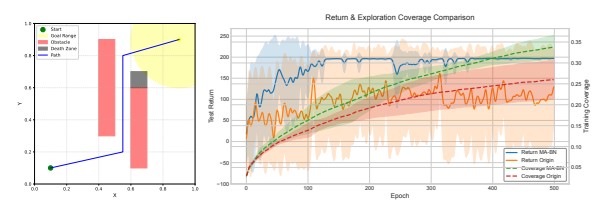

Figure 7: (**Left**) Visualization of the maze environment; (**Right**) Test returns and exploration coverage of DDPG with and without BN (averaged over 5 seeds).

parameters identical. We tracked both test returns and map exploration coverage throughout training. As shown in Figure 7 (right), MA-BN not only accelerates convergence but also significantly increases exploration coverage and the final map coverage. These results suggest that the stochasticity introduced by BN may have the potential to facilitate exploration. By introducing variability into Q-value estimates and action selection, BN may help the agent visit a more diverse set of states, discover better strategies, and reduce the risk of premature convergence to suboptimal policies. Such potential benefits for exploration may be particularly relevant in large or complex RL tasks and could warrant further investigation in future work.

### 4.3 BN Extends the Suitable Learning Rate Range

BN smooths the loss landscape and stabilizes gradients (Santurkar et al., 2018), enabling a wider range of learning rates. To validate this, we ran DRQv2 experiments on DMC tasks. As shown in Figure 8 (left), agents without BN suffer significant performance drops at higher learning rates, whereas MA-BN maintains high returns. These results indicate that MA-BN improves learning-rate robustness, easing training and enhancing convergence.

### 4.4 Batch Normalization Mitigates Optimization Challenges

The growing scale of models amplifies the memory demands of adaptive optimizers such as Adam, which store additional statistics like momentum and variance. This challenge is particularly acute in large models, where memory efficiency is critical. By smoothing the loss landscape and stabilizing gradients, BN reduces optimization difficulty and enables effective training with simpler, memory-efficient optimizers like SGD (Robbins & Monro, 1951). To evaluate this, we compare MA-BN, LN, and Origin across a range of learning rates on DMC tasks. As shown in Figure 8 (right), SGD is generally more sensitive to learning

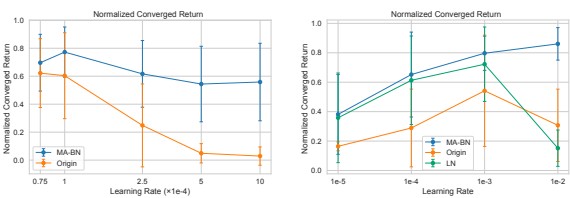

Figure 8: Normalized convergence returns of DRQv2 trained under different normalization methods and learning rates, evaluated on hopper_hop, quadruped_run, and quadruped_walk. **(Left)** under the Adam optimizer with varying learning rates, MA-BN exhibits smaller variability and consistently better convergence compared with Origin. **(Right)** under the SGD optimizer with varying learning rates, MA-BN achieves superior performance relative to both Origin and LN.

rate selection, yet MA-BN markedly alleviates this sensitivity, yielding robust convergence and superior returns across a wide range of rates. These findings indicate that MA-BN can further facilitate optimization and render memory-efficient optimizers such as SGD a practical and effective option, which is particularly valuable in large-scale models.

## 5 Related Work

Batch Normalization and its variants such as Layer Normalization and Group Normalization (GN) have been widely adopted across various deep learning domains. However, their effectiveness in DRL remains underexplored. One of the earliest uses of BN in DRL appears in DDPG, where BN is applied to normalize low-dimensional feature observations with varying physical units and scales across environments. Cobbe et al. (2019) further applied BN to each convolutional layer in the IM-PALA CNN (Espeholt et al., 2018), demonstrating improved generalization. Bhatt et al. (Bhatt et al., 2019) identified the challenge of selecting appropriate BN modes for target networks and proposed CrossNorm, which addresses this issue by mixing on-policy and off-policy samples. However, their work did not provide a systematic analysis of BN mode choices in the actor and critic networks. In contrast, our MA-BN attains superior performance through a principled yet simpler configuration of BN modes, without requiring the mixing samples. Building upon this line of research, Bhatt et al. further proposed the CrossQ algorithm (Bhatt et al., 2024), which eliminates the target network, employs BRN, and adopts a wider critic architecture. These design choices collectively led to marked improvements in both convergence speed and final policy performance, establishing one of the first successful applications of BN variants in DRL. More recently, Palenicek (Palenicek et al., 2025) reported that CrossQ exhibits instability under high update-to-data (UTD) ratios and proposed weight normalization as a remedy, further enhancing training stability. However, CrossQ relies on BRN, and previous works do not provide a systematic analysis of the rationale behind BN mode selection. In contrast, our study is based on canonical BN and offers a comprehensive empirical investigation of different mode choices, yielding practical insights into their impact on off-policy actor-critic frameworks. The mode adopted by CrossQ also aligns with our findings, providing a principled explanation for its design choice.

## 6 CONCLUSION

In this work, we show that appropriately applied Batch Normalization can significantly improve reinforcement learning performance. Through a systematic empirical study in off-policy actor-critic frameworks, we identify failure modes, analyze their causes, and we further propose the MA-BN method based on mode selection guidelines. Our results demonstrate that MA-BN accelerates training, expands the effective learning rate range, enhances exploration, and mitigates optimization difficulties. These findings provide actionable guidance for effectively integrating BN in future RL research and applications.

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

APPENDIX

## A    CRITIC-I MODE SELECTION UNDER ALTERNATIVE CONFIGURATIONS

We investigated the difference between Critic-I-T and Critic-I-E under various BN placements (BN applied before the linear layer) and algorithms. As shown in Table 1, Critic-I-E consistently outperforms Critic-I-T across various environments. To further verify the generality of this observation, we conducted additional experiments using the DDPG algorithm (Lillicrap et al., 2015) in the Pendulum environment, where the same performance gap between Critic-I-T and Critic-I-T was observed. These results suggest that Critic-I E leads to more stable or effective policy learning, regardless of specific BN placement or algorithm.

Table 1: Comparison of converged returns under different modes across various environments and algorithms.

| Algorithm | Environment | Critic-I T | Critic-I E |
|---|---|---|---|
| DRQv2 | hopper_hop | $77.02 \pm 19.73$ | **$278.55 \pm 154.46$** |
| | quadruped_run | $116.54 \pm 65.19$ | **$642.40 \pm 79.44$** |
| | quadruped_walk | $133.98 \pm 68.55$ | **$890.86 \pm 31.63$** |
| DDPG | Pendulum | $-726.85 \pm 69.09$ | **$-393.66 \pm 154.22$** |

## B    PROOF OF THEOREM 1

*Proof.* We first consider the bounded variation of actor parameters over time. Since the update dynamics satisfy $\|\mu_t - \mu_{t+1}\|_2 \leq \Delta$ and $|\delta_t^2 - \delta_{t+1}^2| \leq \Delta$ for all $t$, the pairwise distances between any two means and variances can be bounded by summing over adjacent steps. For any $i < j$,

$$\|\mu_i - \mu_j\|_2 \leq \sum_{k=i}^{j-1} \|\mu_k - \mu_{k+1}\|_2 \leq (j-i)\Delta \leq N\Delta,$$

$$|\delta_i^2 - \delta_j^2| \leq \sum_{k=i}^{j-1} |\delta_k^2 - \delta_{k+1}^2| \leq (j-i)\Delta \leq N\Delta.$$

Next, consider the distribution of actions in the replay buffer. Since each action is sampled from a Gaussian distribution $\mathcal{N}(\mu_t, \delta_t^2)$ at some time step $t$, the entire buffer forms a uniform mixture of $N$ such Gaussians. The mean of a sample drawn from this mixture is:

$$\mathbb{E}[a] = \frac{1}{N} \sum_{t=1}^{N} \mu_t \triangleq \mu_a.$$

The variance of a mixture distribution can be derived from its definition as: $\sigma^2 = \mathbb{E}[a^2] - (\mathbb{E}[a])^2$. The second moment for component $t$ is given by: $\mathbb{E}[a_t^2] = \delta_t^2 + \|\mu_t\|_2^2$. Assuming the mixture consists of $N$ components with uniform weights, the overall second moment becomes:

$$\mathbb{E}[a^2] = \frac{1}{N} \sum_{t=1}^{N} \left( \delta_t^2 + \|\mu_t\|_2^2 \right) = \frac{1}{N} \sum_{t=1}^{N} \delta_t^2 + \frac{1}{N} \sum_{t=1}^{N} \|\mu_t\|_2^2.$$

The square of the mean of the mixture is:

$$(\mathbb{E}[a])^2 = \left\| \frac{1}{N} \sum_{t=1}^{N} \mu_t \right\|_2^2 = \|\mu_a\|_2^2,$$

Substituting the expressions above into the variance formula yields:

$$\sigma_a^2 = \frac{1}{N} \sum_{t=1}^{N} \delta_t^2 + \frac{1}{N} \sum_{t=1}^{N} \|\mu_t\|_2^2 - \|\mu_a\|_2^2.$$

This can be further simplified by invoking the standard identity for the total variance of vectors:

$$\sum_{t=1}^{N} \|\mu_t - \mu_a\|_2^2 = \sum_{t=1}^{N} \|\mu_t\|_2^2 - N\|\mu_a\|_2^2.$$

Applying this identity, we obtain the final expression for the variance:

$$\sigma_a^2 = \frac{1}{N} \sum_{t=1}^{N} \delta_t^2 + \frac{1}{N} \sum_{t=1}^{N} \|\mu_t - \mu_a\|_2^2.$$

Given a batch of $B$ i.i.d. action samples $\{a_1, a_2, \ldots, a_B\}$, where each $a_j$ has mean $\mu_a$ and variance $\sigma_a^2$, the batch mean is defined as $\bar{a} = \frac{1}{B} \sum_{j=1}^{B} a_j$. By standard properties of i.i.d. variables,

$$\mathbb{E}[\bar{a}] = \mu_a, \quad \mathrm{Var}(\bar{a}) = \frac{\sigma_a^2}{B}.$$

Then we complete the proof of theorem 1. $\qquad\square$

## C   DETAILS OF THE BUFFER MIXING EXPERIMENT

We visualize the discrepancy between the action distribution in the replay buffer and that of the current actor policy throughout training in the quadruped_walk environment, as shown in Fig. 9. To investigate the source of distributional mismatch in training, we measure the discrepancy between the action distribution in Step Critic-I and the action distribution stored in the replay buffer. Specifically, we use a randomly sampled batch from the buffer to estimate the latter. Two metrics are computed: the $\ell_2$ norm of the difference in mean and variance between the buffer action $a_b$ and the current policy action with exploration noise $a_c$ (used for calculating batch normalization statistics in Step Critic-I), denoted as $a\_mean\_diff$ and $a\_var\_diff$, respectively.

$$\texttt{a\_mean\_diff} = \|\mathbb{E}[a_c] - \mathbb{E}[a_b]\|_2 \tag{7}$$
$$\texttt{a\_var\_diff} = \|\mathrm{Var}(a_c) - \mathrm{Var}(a_b)\|_2 \tag{8}$$

As shown in Figure. 10, the discrepancy between the two distributions decreases as the proportion of buffer-mixed data increases in most cases. This reduction in distributional gap correlates with improved training stability and convergence, further supporting our hypothesis regarding the critical role of consistent data distribution in Batch Normalization.

We also report the complete training curves and the mean and variance of Q-value estimation bias under different data mixing ratios. As illustrated in Figure 11, both the convergence behavior and estimation accuracy improve as the proportion of buffer-mixed data increases.

## D   CRITIC-I MODE SELECTION IN THE CROSSQ ALGORITHM

We also evaluated the CrossQ algorithm in the MuJoCo environments under different mode selection strategies in Step Critic-I, as shown in the figure 12. Unlike standard architectures, CrossQ removes the target network and trains the critic using mixed data consisting of actions sampled from both the current actor policy and the replay buffer. When operating as Critic-I T, the critic receives inputs only from the current actor's action distribution, which significantly deviates from the mixed distribution

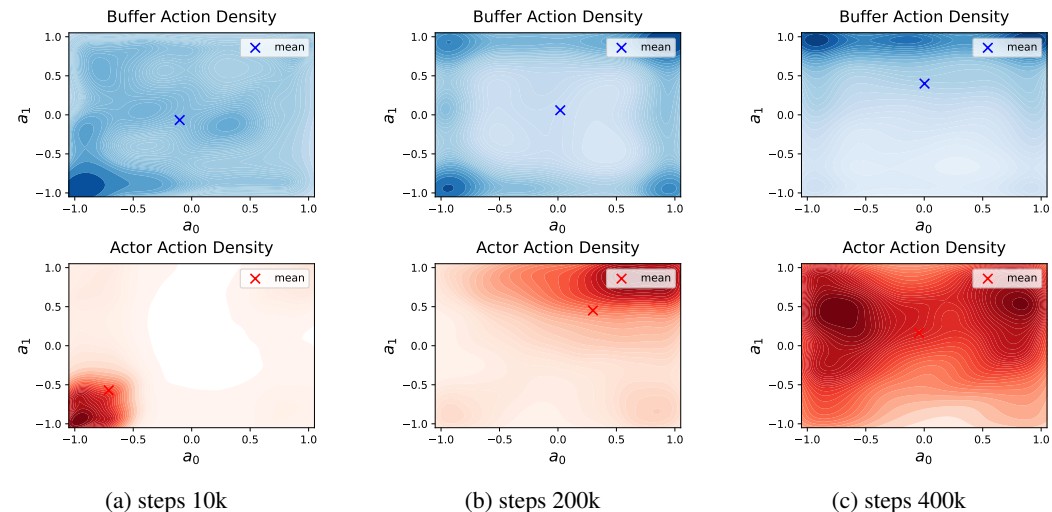

(a) steps 10k                (b) steps 200k                (c) steps 400k

Figure 9: Discrepancy Between Replay Buffer and Current Policy Action Distributions Across Training Steps in quadruped_walk.

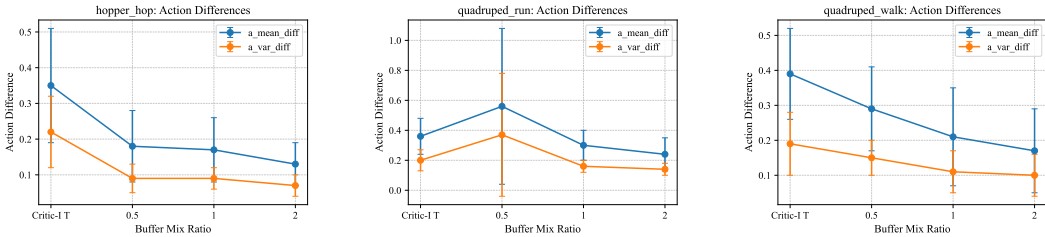

Figure 10: Comparison of action differences under different buffer mixing ratios across environments. From left to right: hopper_hop, quadruped_run, quadruped_walk.

seen during training. This distribution mismatch leads to highly unstable training behavior, often resulting in divergence or NaN values. In contrast, Critic-I E yields stable training under the same conditions. Furthermore, when actions from the replay buffer are still mixed in training mode during Step Critic-I (denoted as Critic-I mixdata), the resulting distribution aligns with that of the training phase. In this case, actor learning remains stable and Q-value estimates are accurate, allowing the algorithm to converge successfully. These results further support our earlier hypothesis that distribution mismatch is the primary cause of training instability under the training mode in Step Critic-I.

## E  CRITIC-I MODE SELECTION UNDER DISCRETE ACTIONS

We further investigate the impact of different mode selection strategies in Step Critic-I of the SD-SAC algorithm (Zhou et al., 2022) under discrete action settings by evaluating its performance on three Atari environments (Bellemare et al., 2013): Pong, Alien, and Freeway. Unlike methods that rely on sampling a single action, SD-SAC computes the expected state-action value over the entire action space, mitigating discrepancies between the action distributions during training and evaluation. As a result, it is able to accurately estimate and update the Q-values in both modes, leading to comparable final performance across settings, as illustrated in Figure 13.

## F  A TOY EXAMPLE ON THE NEGATIVE IMPACT OF TRAINING MODE

To investigate the potential negative effects of using BN in training mode during policy evaluation, we consider a simple discrete-time Linear Quadratic Regulator (LQR) problem with scalar state and

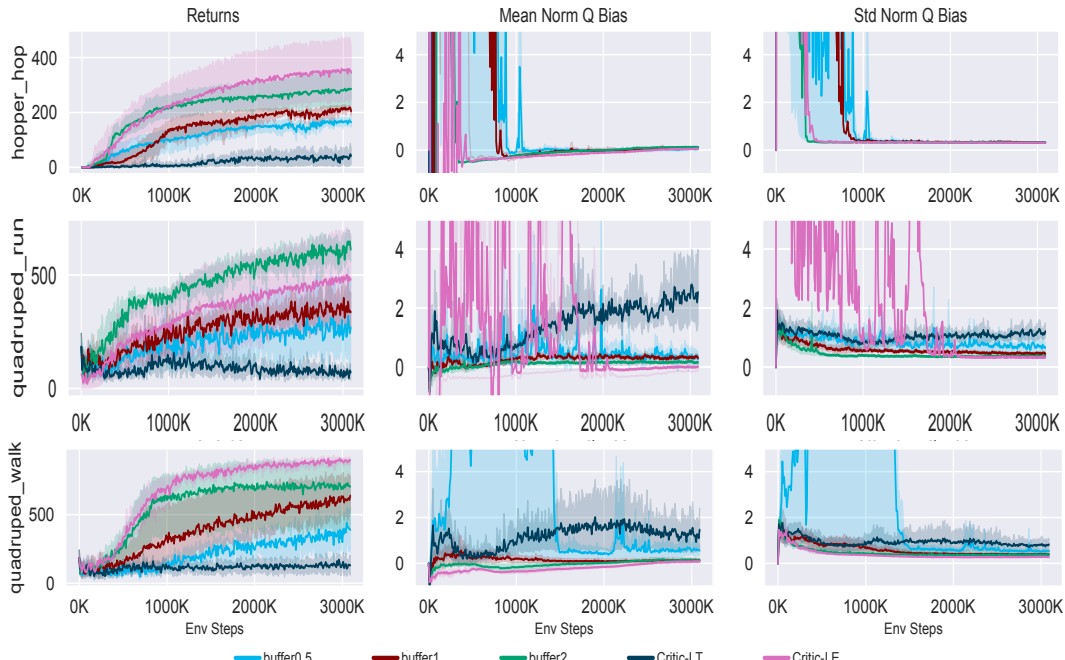

Figure 11: Performance curve comparison of DRQv2 under different buffer mixing ratios in training mode.

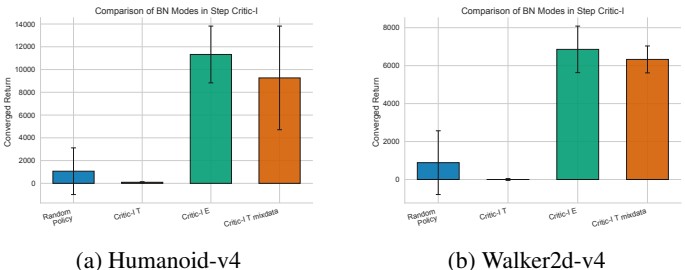

(a) Humanoid-v4         (b) Walker2d-v4

Figure 12: Comparison of converged returns of the CrossQ algorithm across different mode selection strategies in various Mujoco environments.

action. The system dynamics are given by:

$$s_{t+1} = As_t + Ba_t, \tag{9}$$

where $s_t \in \mathbb{R}$ is the state and $a_t \in \mathbb{R}$ is the control input. The objective is to minimize the infinite-horizon discounted cost:

$$J = \sum_{t=0}^{\infty} \gamma^t \left( Qs_t^2 + Ra_t^2 \right), \tag{10}$$

with scalar weights $Q > 0$ and $R > 0$. The optimal value function is quadratic: $V^*(s) = Ps^2$, where $P$ is the unique positive solution to the scalar Riccati equation:

$$aP^2 + bP + c = 0, \tag{11}$$

with

$$a = \gamma B^2, \quad b = R(1 - \gamma A^2) - \gamma QB^2, \quad c = -QR, \tag{12}$$

and closed-form solution:

$$P = \frac{-b + \sqrt{b^2 - 4ac}}{2a}. \tag{13}$$

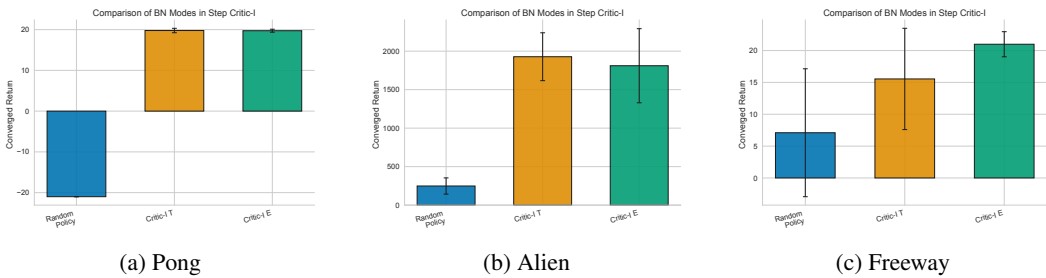

(a) Pong           (b) Alien           (c) Freeway

Figure 13: Comparison of converged returns of the SD-SAC algorithm across different mode selection strategies in various Atari environments.

Using $P$, the optimal Q-function admits the closed-form expression:

$$Q^*(s,a) = -\left(Q_s s^2 + R_a a^2 + 2Nsa\right),\tag{14}$$

where

$$Q_s = Q + \gamma A^2 P, \quad R_a = R + \gamma B^2 P, \quad N = \gamma ABP.\tag{15}$$

This analytic form allows precise evaluation of policy and Q-value errors. We train a DDPG agent in an LQR environment, where the critic employs BN and operates with TTT mode. At epoch 2, we visualize the critic's Q-value estimates over the full state-action space in both training and evaluation modes, as shown in Fig. 14. In evaluation mode (middle subplot), the use of running statistics in BN helps mitigate local estimation bias arising from a single mini-batch, resulting in smoother and more consistent Q-value landscapes. In contrast, the training mode (left subplot) relies on batch-specific statistics, which can be biased due to the non-uniform distribution of the sampled state-action pairs. This leads to skewed Q-value estimates that may inadvertently encode the actor's current policy bias. Consequently, the actor is updated based on these biased targets, reinforcing local preferences and generating increasingly biased data. This feedback loop can lead to a self-reinforcing failure mode in which both the actor and critic diverge from optimal behavior. As illustrated in Fig. 15, by epoch 4, the learned policy becomes nearly the inverse of the optimal one. At this stage, the experience buffer itself is heavily biased, such that even switching back to evaluation mode yields inaccurate Q-value estimates due to the shifted data distribution.

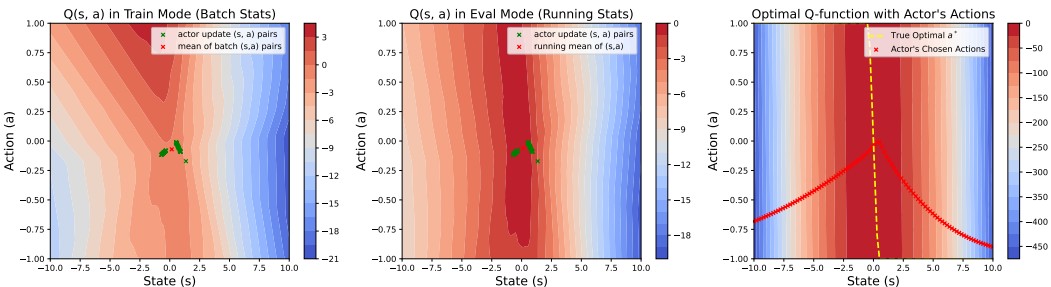

Figure 14: Visualization of actor and critic behavior at epoch 2.

## G   BN MODE SELECTION IN THE ACTOR NETWORK UNDER THE CROSSQ ALGORITHM

As shown in Figure 16, we evaluate the performance of the CrossQ (Bhatt et al., 2024) algorithm on two representative continuous-control tasks from the MuJoCo (Todorov et al., 2012) benchmark under four BRN mode configurations in the actor network: "TT", "TE", "ET", and "Origin". In the `Humanoid-v4` environment, the "Origin" configuration achieves the highest performance, while the other BN-enabled modes introduce additional stochasticity, potentially enhancing exploration and exhibiting larger variance, indicating their potential to discover higher returnss. In the

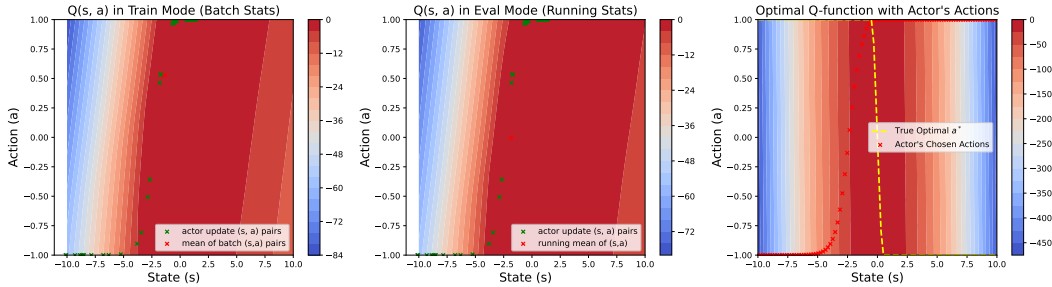

Figure 15: Visualization of actor and critic behavior at epoch 4.

`Walker2d-v4` environment, the "TT" mode performs best. Overall, using BRN with the TT configuration maximizes beneficial stochasticity, potentially leading to improved performance and stable training in certain tasks. This effect may be even more pronounced in complex tasks with deeper network architectures.

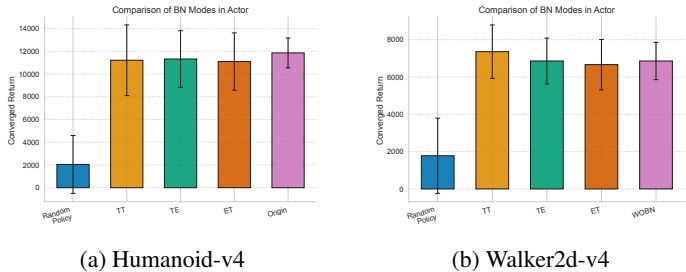

(a) Humanoid-v4      (b) Walker2d-v4

Figure 16: Comparison of converged returns under different BRN configurations in the actor network using the CrossQ algorithm.

## H ADDITIONAL EXPLORATION EXPERIMENTS

To further examine the potential relationship between BN stochasticity and exploration, we conducted experiments using the SAC algorithm in the maze2d-umaze and maze2d-medium environments. We compared three methods, MA-BN, LN, and Origin, under identical hyper-parameter settings, evaluating their coverage of the state space. The results are presented in Figure 17. In the relatively simple umaze task, all three methods successfully completed the task, achieving comparable coverage. In the more challenging medium environment, MA-BN demonstrated higher coverage. This suggests that the stochasticity introduced by batch statistics may facilitate exploration, allowing the agent to traverse the environment more thoroughly. Importantly, this mechanism is orthogonal to conventional exploration strategies in reinforcement learning, such as entropy-based methods (Tiapkin et al., 2023) or curiosity-driven methods (Li et al., 2020), and can serve as a complementary approach to further enhance exploration.

## I HYPER-PARAMETER SETTINGS

We adopt the default hyper-parameter settings for Drqv2, as listed in Table 2. For the CrossQ and SAC algorithms, we follow the default settings reported in the original CrossQ paper, summarized in Table 3. The SD-SAC algorithm uses the default hyper-parameters specified in its original implementation, as shown in Table 4. Finally, for the DDPG algorithm, we adopt the default settings provided by the WiseRL repository, summarized in Table 5.

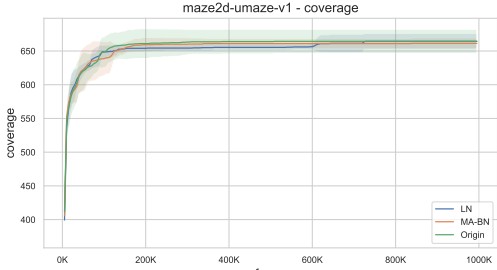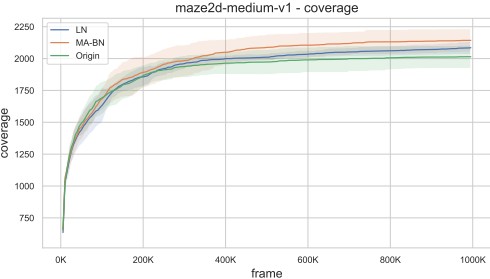

Figure 17: Evaluation of state-space coverage for MA-BN, LN, and the original baseline in the maze2d-umaze and maze2d-medium environments, averaged over three random seeds.

Table 2: Hyper-parameters used in our DrQv2 experiments.

| Parameter | Setting |
|---|---|
| Replay buffer capacity | $10^6$ (or $10^5$ for `quadruped_run`) |
| Exploration steps | 2000 |
| Mini-batch size | 256 |
| Discount factor $\gamma$ | 0.99 |
| Optimizer | Adam |
| Learning rate | $10^{-4}$ |
| Agent update frequency | 2 |
| Critic Q-function soft-update rate $\tau$ | 0.01 |
| Hidden dimension | 1024 |
| Exploration stddev. clip | 0.3 |

## J   USE OF LARGE LANGUAGE MODELS

Some portions of the text were polished with the assistance of Large Language Models (LLMs). All content remains the responsibility of the authors.

Table 3: Hyper-parameters for SAC and CrossQ.

| Parameter | SAC | CrossQ |
|---|---|---|
| Discount Factor ($\gamma$) | 0.99 | |
| Learning Rate (Actor & Critic) | 0.001 | |
| Replay Buffer Size | $10^6$ | |
| Batch Size | 256 | |
| Critic Width | 256 | 2048 |
| Target Update Rate ($\tau$) | 0.005 | N/A |
| Adam $\beta_1$ | 0.9 | 0.5 |

Table 4: Hyper-parameters for SD-SAC.

| Hyperparameter | Setting |
|---|---|
| Learning rate | $10^{-5}$ |
| Optimizer | Adam |
| Mini-batch size | 64 |
| Discount factor ($\gamma$) | 0.99 |
| Replay buffer size | $10^5$ |
| Number of hidden layers | 2 |
| Hidden units per layer | 512 |
| Target smoothing coefficient ($\tau$) | 0.005 |
| Alpha | 0.05 |
| Beta | 0.5 |
| c | 0.5 |

Table 5: Hyper-parameters for DDPG.

| Hyperparameter | Setting |
|---|---|
| Hidden layer dimensions | (256, 128) |
| Batch size | 64 |
| Replay buffer size | 1000 |
| Learning rate (actor) | $3 \times 10^{-4}$ |
| Optimizer | Adam |
| Discount factor ($\gamma$) | 0.99 |
| Target smoothing coefficient ($\tau$) | 0.005 |

