# OpenReview forum: "An Investigation of Batch Normalization in Off-Policy Actor-Critic Algorithms"
_ICLR.cc/2026/Conference — Submitted to ICLR 2026_

### Official Review · Reviewer_my1n · 2025-11-01

**Soundness:** 2
**Presentation:** 2
**Contribution:** 2
**Rating:** 4
**Confidence:** 3

**Summary:**

This paper investigates the effectiveness of applying Batch Normalization (BN) to Deep Reinforcement Learning (DRL), particularly within off-policy actor-critic (AC) frameworks. While BN is a well-known technique for stabilizing the training of deep learning models, its applicability to DRL is not straightforward due to the nature of the IID data from the replay buffer.

This paper empirically investigates a study focusing on a critical, yet underexplored, aspect: the distinct behaviors of BN's training mode versus its evaluation mode at different stages of the AC update.
The authors found the failure modes that collapse the stability and performance, for example, when BN is used in in training mode during the actor update (Critic-I step).

Based on this observation, the paper proposes Mode-Aware Batch Normalization (MA-BN), a set of simple, practical guidelines for setting the BN modes. They demonstrate empirically that MA-BN not only stabilizes training but also outperforms Layer Normalization (LN) and no-normalization baselines, accelerates convergence, and enhances exploration.

**Strengths:**

This paper provides a method for decomposing the problem of why BN fails and succeeds in DRL, especially in AC settings.

1) The notation is clear and easy to understand. They point out cases where BNs are used as Actor I and II, and Critic I, II, and III as described in Algorithm 1.

2) The analysis is comprehensive, covering multiple algorithms (DRQv2, SAC, DDPG, SD-SAC), benchmarks (DMC, Mujoco, Atari), and even related work (CrossQ in Appendix D).

3) The paper demystifies why BN has been so troublesome in RL, providing a concrete, actionable diagnosis.

4) It provides an immediate, simple, and "drop-in" solution (the MA-BN "ETT" critic config) for practitioners.

**Weaknesses:**

Although the paper explores the BNs in AC settings comprehensively, the justifications for the final MA-BN configuration are not uniformly rigorous. The analysis for Critic-I (Section 3.1) is conclusive, but the analyses for Critic-III and the Actor (3.2 and 3.3) are less definitive.

1) Justification for Critic-III T (Target Critic)

The analysis in Section 3.2 (Fig. 4) shows that ETT is strong, but ETE (eval mode) with soft-updated BN stats ("ETE+BN-soft") is also competitive and even superior in one environment. The paper's choice of ETT for "simplicity" feels slightly arbitrary, as "ETE+BN-soft" is arguably just as simple and perhaps more intuitive.

2) Justification for Actor TT
The analysis for the actor mode (Section 3.3, Fig. 5) is also inconclusive. "TT" is chosen as the default, but "ET" performs best for SAC on Mujoco. The authors justify "TT" as "maximizing stochasticity, but the results imply the optimal actor mode may be algorithm- or environment-dependent.

**Questions:**

1) Actor choice
The choice of Actor TT seems based on good performance on DMC, even though ET was best for SAC on Mujoco (Fig 5). This suggests the optimal actor mode might be algorithm-dependent. How critical is the TT choice for the DRQv2 results in Section 4? Would performance degrade significantly if you used ET instead?

2) Figure clarity
To help readers better understand the paper, I would suggest adding more explanation of the results in the figure captions. For example, the connection between BN effectiveness and the experiment and results in Section 4.4. (Figure 8) seems unclear.

3) Method applicability
In this paper, the effectiveness of BN is investigated in online off-policy AC RL settings. Are the insights in the paper applicable to offline RL settings, where the estimation of values is more important and more complex due to the biased distribution between the dataset and the environment?

---

> ### Author Response · Authors · 2025-11-20
> **Response to Reviewer #my1n**
>
> We thank Reviewer my1n for highlighting the clear notation, comprehensive analysis across algorithms and benchmarks, the actionable diagnosis of BN behavior in RL, and the practical value of our MA-BN solution. A detailed response is provided below.
>
> > w1. Justification for Critic-III T (Target Critic) The analysis in Section 3.2 (Fig. 4) shows that ETT is strong, but ETE (eval mode) with soft-updated BN stats ("ETE+BN-soft") is also competitive and even superior in one environment. The paper's choice of ETT for "simplicity" feels slightly arbitrary, as "ETE+BN-soft" is arguably just as simple and perhaps more intuitive.
>
> We thank the reviewer for the comment. Due to the high complexity of neural networks, the magnitude of parameter updates and the corresponding updates of the batch normalization statistics may exhibit nonlinear relationships. Consequently, in the case of ETE+BN-soft, this mismatch between the updates of the BN statistics and the network parameters may degrade performance. In contrast, ETT operates in training mode, which circumvents this issue while still achieving strong performance.
>
> > w2. Justification for Actor TT The analysis for the actor mode (Section 3.3, Fig. 5) is also inconclusive. "TT" is chosen as the default, but "ET" performs best for SAC on Mujoco. The authors justify "TT" as "maximizing stochasticity, but the results imply the optimal actor mode may be algorithm- or environment-dependent.
>
> We thank the reviewer for the comment. Regarding the choice of ET versus TT modes for the Actor network, the overall impact on performance is minimal, as discussed in response to q1. Nevertheless, TT maximizes exploration and is therefore adopted as the default choice.
>
> > q1: Actor choice The choice of Actor TT seems based on good performance on DMC, even though ET was best for SAC on Mujoco (Fig 5). This suggests the optimal actor mode might be algorithm-dependent. How critical is the TT choice for the DRQv2 results in Section 4? Would performance degrade significantly if you used ET instead?
>
> Thank you for your question. Regarding the mode selection of the actor network, our main conclusion is that it has only a minor influence on overall performance. The TT mode introduces the highest degree of stochasticity, and we therefore adopt it as the default configuration. We additionally evaluate the ET setting on two tasks under the DrQv2 algorithm (averaged over 5 seeds), and the coverage return are summarized below. Overall, its impact on performance is minor.
>
> | Actor Mode | quadruped_run   | quadruped_walk |
> | ---------- | --------------- | -------------- |
> | ET         | 515.90 ± 187.77 | 887.39 ± 31.08 |
> | TT         | 482.03 ± 238.82 | 890.86 ± 31.63 |
>
>
> > q2: Figure clarity To help readers better understand the paper, I would suggest adding more explanation of the results in the figure captions. For example, the connection between BN effectiveness and the experiment and results in Section 4.4. (Figure 8) seems unclear.
>
> Thank you for the suggestion. In the revised manuscript, we have added a detailed explanation of Figure 8 to improve clarity and overall readability.

---

> ### Author Response · Authors · 2025-11-20
> **Response to Reviewer #my1n**
>
> > q3: Method applicability In this paper, the effectiveness of BN is investigated in online off-policy AC RL settings. Are the insights in the paper applicable to offline RL settings, where the estimation of values is more important and more complex due to the biased distribution between the dataset and the environment?
>
> Thank you for the question. Our work focuses on online off-policy actor-critic algorithms, where the continual interaction inherent to reinforcement learning induces distribution shift and thereby poses nontrivial challenges for the use of batch normalization. Our analysis shows that, when applied correctly, batch normalization can nevertheless support stable training and provide advantages in parameter robustness, training stability, and exploration. In offline reinforcement learning, by contrast, the absence of interaction-driven distribution shift makes the training process more similar to supervised learning, a domain in which batch normalization has already demonstrated substantial effectiveness[1]. Consequently, batch normalization has the potential to stabilize and accelerate offline RL training, and its adaptive statistics may further enable rapid domain adaptation[2, 3] under distributional mismatch. As this paper primarily investigates the online setting, a comprehensive study of batch normalization in offline RL is left for future work.
>
> [1] Santurkar S, Tsipras D, Ilyas A, et al. How does batch normalization help optimization?[J]. Advances in neural information processing systems, 2018, 31.
>
> [2] Wang D, Shelhamer E, Liu S, et al. Tent: Fully test-time adaptation by entropy minimization[J]. arXiv preprint arXiv:2006.10726, 2020.
>
> [3] Schneider S, Rusak E, Eck L, et al. Improving robustness against common corruptions by covariate shift adaptation[J]. Advances in neural information processing systems, 2020, 33: 11539-11551.

---

### Official Review · Reviewer_73gA · 2025-11-01

**Soundness:** 3
**Presentation:** 3
**Contribution:** 2
**Rating:** 2
**Confidence:** 4

**Summary:**

The paper explores how Batch Normalization (BN) behaves within deep reinforcement learning in off-policy actor-critic methods. Through empirical analysis, authors show that BN can still offer benefits when configured correctly. They uncover that improper selection between BN’s training and evaluation modes can lead to instability or divergence, and propose Mode-Aware Batch Normalization (MA-BN), as the best configuration. Authors show that MA-BN can improve exploration and hyperparameters stability.

**Strengths:**

Putting aside the motivation issues described later, the paper is technically well written. All experiments are clearly explained, and the figures actually demonstrate the phenomena being studied. I have no remarks on the design of the existing experiments. The only thing missing is the details necessary for reproducibility. I advise the authors to explicitly add all hyperparameters and describe the setup of each experiment in detail. Currently, this is not the case.

**Weaknesses:**

Overall, the paper is currently more like a collection of unrelated experiments without clear single motivation and lacks clear conclusive experiments.

If the goal is to demonstrate the advantages of BN over other normalization methods, more extensive comparisons with state-of-the-art algorithms are required (at least on the level of modern algorithms, like SimBa, BroNet, etc). Judging by the results in Figure 6, even the best found BN configuration doest not outperform LN. While authors do not claim state-of-the-art, this results undermines the inital motivation that BN is worth such attention at all. If LN works just fine, why bother? I think paper does not address this question convincingly (in contrast to CrossQ which focuses on simplification and faster training).

If the goal is to analyse unique properties that BN can give in comparison to other methods (like mentioned improved exploration), a more in-depth and comprehensive analysis is needed, including not only LN but other methods of regularization and exploration. I liked the experiments on improving exploration and stability to hyperparameters the most. These results, with more detailed development, may actually be novel and could be useful and interesting. However, very little attention is paid to them. Exploration studied in over simplified setup, which may not transfer to the more difficult and stochastic environment (good similar example is the recent study on the effect of small batch size on exploration, see https://arxiv.org/abs/2310.03882). What other properties make BN unique in comparison to the other methods? Such results would be significantly more insightful.

As for the main analysis regarding the BN modes, it is not convincing. If I understand results correctly, the authors are essentially reproduced CrossQ's conclusions for Critic 1. Moreover, authors write: “… reinforcing our claim that distribution mismatch is a key factor behind the degradation of training-mode performance”. Why this claim and it’s confirmation is novel, given the CrossQ results? Seems like CrossQ addressed this problem specifically and with similar methods before. As for the rest of the analysis of Critic 2-3, it is completely unnecessary in Cross-Q as it removes target networks (which is a good thing which we must strive for anyway). Given that CrossQ is already quite old (first appeared in 2019), and stood the test of time, I do not understand the motivation to study BN based on DRQ and not CrossQ. In conclusion, I believe that changing one hyperparameter is not sufficient for a separate method name.

Overall, reading the paper gives a mixed impression. To summarise, I feel that despite the (in my opinion) complete technical validity of the results, they are insufficiently significant and novel for publication at ICLR, and it would be more suitable for publication in for example TMRL, which explicitly values technical correctness over the possible impact.

**Questions:**

1. What are the advantages/motivation on using DRQ instead of Cross-Q?
2. Given the rising need for continual learning, do you expect that BN will perform on the same level in the streaming setups? (see https://arxiv.org/abs/2410.14606 or https://arxiv.org/abs/2411.15370)

---

> ### Author Response · Authors · 2025-11-20
> **Response to Reviewer #73gA**
>
> We thank Reviewer 73gA for recognizing the technical clarity, well-designed experiments, and informative figures, and we will address the reproducibility concerns by providing full hyperparameter and setup details below.
>
> weakness: Thank you for your comment. Regarding reproducibility, we provide experimental parameter settings in the updated manuscript and included an anonymous code repository in the initial submission to ensure reproducibility. We would like to reiterate our motivation: our main argument is that, in online off-policy actor-critic algorithms, batch normalization has the potential to outperform layer normalization when used correctly. While layer normalization has been more commonly adopted in the reinforcement learning community, batch normalization may serve as a superior alternative under appropriate usage. To investigate correct usage, we first selected the DrQv2 algorithm, which is representative of many off-policy actor-critic methods, and analyzed batch normalization mode selection during training. Our study also covers SAC, DDPG, CrossQ, SD-SAC, both continuous and discrete action spaces, pixel- and state-based observations, and multiple benchmarks to ensure the generality of our analysis. Through this systematic study, we provide practical guidance on mode selection and analyze the failure of specific mode combinations to derive recommendations for proper usage. Building on this, we show in Figure 6 that batch normalization achieves superior performance to layer normalization in four scenarios and comparable performance in three scenarios. Beyond convergence performance, we demonstrate that batch normalization exhibits stronger robustness to learning rates, enabling successful training even with the SGD optimizer while outperforming layer normalization in terms of return. This is particularly significant in large-scale model settings, where using lighter optimizers such as SGD can save memory. Furthermore, the inherent batch-wise stochasticity of batch normalization introduces randomness that can enhance exploration, which is fully orthogonal to existing mechanisms such as curiosity or entropy-based exploration, providing an additional complementary approach. Additional SAC exploration experiments on the Umaze tasks are provided in Appendix H of the revised manuscript. While the CrossQ algorithm achieves state-of-the-art performance in continuous-control Mujoco tasks, it does not explain the rationale behind its mode selection and does not cover a broader range of benchmarks, continuous and discrete action spaces, or pixel- and state-based observations. Our work provides a general analysis of off-policy actor–critic algorithms in the online setting, covering methods such as DDPG, SAC, and SD-SAC, with CrossQ being only a special case within this broader framework. The mode selection used in CrossQ is consistent with our conclusions, in particular Critic I operates in E mode, although the original paper does not explicitly discuss this choice. We have verified the mode configuration in the official CrossQ implementation, and the results agree with our analysis; see Appendices D and G for details. We further offer a more generalizable study, highlighting the advantages of batch normalization over layer normalization in terms of learning rate sensitivity, training difficulty, exploration, and convergence.

---

> ### Author Response · Authors · 2025-11-20
> **Response to Reviewer #73gA**
>
> > q1: What are the advantages/motivation on using DRQ instead of Cross-Q?
>
> Thank you for your question. Our study analyzes off-policy actor-critic algorithms in the online setting. We selected the DrQv2 algorithm due to its more general training workflow, which includes comprehensive steps from actor-I and actor-II to critic-I through critic-III. This design allows it to effectively encompass many off-policy actor-critic algorithms, such as SAC, DDPG, and SD-SAC for discrete action spaces. Our study covers a wide range of off-policy actor-critic algorithms across both continuous and discrete action spaces, pixel- and state-based observations, and multiple benchmark environments. In contrast, CrossQ, as a specialized improvement of an off-policy actor-critic algorithm, removes the target network and thus lacks a generalizable workflow. Using CrossQ as the primary analysis framework would limit the applicability of our findings to other off-policy actor-critic algorithms, which contradicts the purpose of our study. Moreover, our original manuscript provides an analysis of mode selection for CrossQ's Critic-I in Appendix D and G, addressing aspects that were not examined in the CrossQ paper itself.
>
>
> > q2: Given the rising need for continual learning, do you expect that BN will perform on the same level in the streaming setups? (see https://arxiv.org/abs/2410.14606 or https://arxiv.org/abs/2411.15370)
>
> Thank you for your question. Our study primarily focuses on off-policy actor-critic algorithms in the online setting. Under a streaming scenario, where only a single sample is available at each step, batch normalization is not suitable for performing meaningful normalization, as it is designed for batch-wise statistics. Our main conclusion is that, when used correctly in online off-policy actor-critic algorithms, batch normalization has the potential to outperform layer normalization due to its parameter robustness, the exploration benefits induced by batch-wise normalization, ease of training, and superior performance. Analyses under other settings are left for future work, as they fall outside the scope of the problem addressed in this paper.

---

### Official Review · Reviewer_Fd95 · 2025-11-01

**Soundness:** 2
**Presentation:** 2
**Contribution:** 1
**Rating:** 2
**Confidence:** 4

**Summary:**

This paper examines the use of Batch Normalization (BN) in off-policy actor-critic reinforcement learning, with a focus on how different BN modes - training vs. evaluation - affect learning dynamics. The authors propose Mode-Aware Batch Normalization (MA-BN) as a solution to observed instability issues, claiming that it enhances exploration and improves robustness to hyperparameter variation. While the premise might appear promising, the overall execution and depth of analysis fall short of making a meaningful contribution.

**Strengths:**

The paper is technically competent and clearly structured. Figures are informative and do illustrate the discussed effects. The idea of making BN mode selection adaptive (MA-BN) is at least conceptually reasonable, and the authors deserve credit for a clean empirical setup. However, beyond these basic merits, the work struggles to justify its necessity or originality.

**Weaknesses:**

The study lacks a coherent purpose and fails to articulate a strong motivation. It is unclear whether the intent is to promote BN as a superior normalization method or to simply catalogue its behavior under certain conditions. The findings largely reiterate what prior work -particularly CrossQ has already shown, yet without acknowledging the redundancy. More importantly, BN does not demonstrate any real advantage over Layer Normalization (LN), making the central claim weak and unconvincing. The exploration experiments, while potentially interesting, are shallow and performed in oversimplified environments that do not reflect the challenges of realistic RL tasks. Overall, the work feels incremental, derivative, and unlikely to influence future research directions.

**Questions:**

(1) Why choose DrQ over CrossQ as the foundation of the analysis? The decision to base the study on DrQ instead of the more relevant CrossQ framework seems arbitrary and poorly justified.

(2) How would BN perform in continual learning setups? BN fundamentally depends on batch-level statistics, which makes it ill-suited for non-stationary or streaming scenarios where data distributions evolve over time. Have the authors considered how MA-BN might behave under distribution drift? Would it require constant re-estimation of running statistics, or would that simply destabilize learning further?

---

> ### Author Response · Authors · 2025-11-20
> **Response to Reviewer #Fd95**
>
> We thank Reviewer Fd95 for acknowledging the technical soundness, clear structure, informative figures, and the conceptual reasonableness of MA-BN, while we address the concerns regarding necessity and originality below.
>
> weakness: Thank you for your comment. We would like to reiterate our motivation: the core contribution of our study is to demonstrate that, under correct usage, batch normalization has the potential to serve as a more effective regularization method than layer normalization in off-policy actor-critic algorithms, even though layer normalization is widely used in RL. To clarify how batch normalization should be properly applied, and to distinguish between training and evaluation modes in reinforcement learning, we first conducted systematic experiments analyzing mode selection at each update step. We further examined the reasons behind the failure of certain mode combinations and provided practical guidance, highlighting the advantages of batch normalization over layer normalization in terms of performance, hyperparameter sensitivity, and training stability.
>
> Unlike CrossQ, which circumvents data distribution mismatch by removing the target network and altering data processing, our study provides a systematic analysis of correct mode selection, identifies the causes of failures under specific combinations, and offers practical recommendations. As a general investigation, we designed experiments across multiple off-policy actor-critic algorithms, including CrossQ, SAC, DDPG, DrQv2, and SD-SAC, covering both continuous and discrete action spaces and multiple benchmarks. Our work provides a general analysis of off-policy actor–critic algorithms in the online setting, covering methods such as DDPG, SAC, and SD-SAC, with CrossQ being only a special case within this broader framework. The mode selection used in CrossQ is consistent with our conclusions, in particular Critic I operates in E mode, although the original paper does not explicitly discuss this choice. We have verified the mode configuration in the official CrossQ implementation, and the results agree with our analysis; see Appendices D and G for details.
>
> Through this comprehensive analysis, we verified that batch normalization achieves comparable or superior performance to layer normalization in many tasks (Fig. 6), exhibits stronger parameter robustness, and is easier to train, even with simple SGD optimizers, which is particularly significant in the context of current large-scale model research. Moreover, the inherent batch-wise stochasticity of batch normalization may facilitate exploration, complementing existing methods such as curiosity or entropy-based exploration. We additionally include SAC exploration experiments in the maze2d environment, reported in Appendix H of the revised manuscript.

---

> ### Author Response · Authors · 2025-11-20
> **Response to Reviewer #Fd95**
>
> > q1 : Why choose DrQ over CrossQ as the foundation of the analysis? The decision to base the study on DrQ instead of the more relevant CrossQ framework seems arbitrary and poorly justified.
>
> Thank you for your question. As a systematic analysis of off-policy actor-critic algorithms, we selected the DrQv2 algorithm due to its more general training workflow, which includes comprehensive steps from actor-I and actor-II to critic-I through critic-III. This design allows it to effectively encompass many off-policy actor-critic algorithms, such as SAC, DDPG, and SD-SAC. Our study covers a wide range of off-policy actor-critic algorithms across both continuous and discrete action spaces, pixel and state observations, and multiple benchmark environments. In contrast, CrossQ, as a specialized improvement of an off-policy actor-critic algorithm, removes the target network and thus lacks a generalizable workflow. Using CrossQ as the primary analysis framework would limit the applicability of our findings to other off-policy actor-critic algorithms, which contradicts the purpose of our study. Moreover, our original manuscript provides an analysis of mode selection for CrossQ's Critic-I in Appendix D and G, which addresses aspects not examined in the CrossQ paper itself.
>
>
> > q2 : How would BN perform in continual learning setups? BN fundamentally depends on batch-level statistics, which makes it ill-suited for non-stationary or streaming scenarios where data distributions evolve over time. Have the authors considered how MA-BN might behave under distribution drift? Would it require constant re-estimation of running statistics, or would that simply destabilize learning further?
>
> Thank you for your question. Our study primarily focuses on off-policy actor-critic algorithms in the online setting. Under a streaming scenario, where only a single sample is available at each step, batch normalization is not suitable for performing meaningful normalization, as it is designed for batch-wise statistics. Our main conclusion is that, when used correctly in online off-policy actor-critic algorithms, batch normalization has the potential to outperform layer normalization due to its parameter robustness, the exploration benefits induced by batch-wise normalization, ease of training, and superior performance. This advantage is particularly notable in large-scale model settings, where the use of lighter optimizers such as SGD is beneficial. Analyses under other settings are left for future work, as they fall outside the scope of the problem addressed in this paper.

---

> ### Comment · Area_Chair_dR4P · 2025-11-25
> **Request from clarification from authors.**
>
> Dear Authors,
>
> In your response to the reviewer, you said that you "verified that batch normalization achieves comparable or superior performance to layer normalisation in many tasks (Fig. 6)". Figure 6 in the paper does not seem to support this claim.
>
> Do you have evidence that BN outperforms LN in a statistically significant way on at least one task?
>
> Thanks,
>
> Area Chair

---

> > ### Author Response · Authors · 2025-11-26
> > **Response to AC**
> >
> > Dear Area Chair,
> >
> > Thank you for your careful reading and for raising this point.
> >
> > Regarding your question: in Figure 6, across the eight evaluated tasks, Batch Normalization (BN) exhibits faster convergence or superior final performance compared to Layer Normalization (LN) on four tasks, and matches LN on two additional tasks, providing empirical support for its effectiveness. Furthermore, Figure 8 shows that when using the SGD optimizer, BN achieves consistently better convergence than LN over a substantial range of learning rates, suggesting that BN may be easier to train and more robust to hyperparameter choices. This property is particularly relevant in the context of large-scale models, where employing lighter-weight optimizers such as SGD can offer significant computational advantages. We hope these clarifications address your concerns.

---

### Official Review · Reviewer_9xBN · 2025-11-03

**Soundness:** 3
**Presentation:** 3
**Contribution:** 3
**Rating:** 6
**Confidence:** 4

**Summary:**

This paper presents a comprehensive empirical investigation of Batch Normalization (BN) in off-policy actor-critic algorithms. The authors analyze all possible BN placement and mode combinations (training vs evaluation) for both the actor and critic networks, identifying configurations that lead to instability or divergence. They find that using BN in evaluation mode for the critic’s actor-update step (Critic-I) avoids a distribution mismatch between the buffer and current policy, while training mode generally works best elsewhere. Based on this study, they propose Mode-Aware Batch Normalization (MA-BN), a principled configuration that mproves stability, accelerates convergence, widens the usable learning-rate range, and modestly enhances exploration.

**Strengths:**

- **Comprehensive and systematic study:** The paper carefully evaluates BN modes at every relevant point in the off-policy actor-critic pipeline, filling an important empirical gap in RL normalization literature.
- **Clarity and motivation:** The motivation is clear. BN is underused in RL due to non-i.i.d. data, and the paper provides a principled exploration of why.
- **Important Insights:** The results lead to a concrete, easy-to-implement recipe (MA-BN) that can directly guide practitioners.
- **Relevance:** Understanding normalization behavior in off-policy RL is increasingly important for scalable visual and continuous-control settings.
- **Presentation:** The paper is generally well-written and easy to follow, with thorough experiments on both continuous and discrete benchmarks.

**Weaknesses:**

1. **Terminology:** The use of “training” and “evaluation” modes for BN may confuse readers, since both modes are used during RL training. Terms like *instantaneous BN* (using batch statistics) vs *running-average BN* might reduce ambiguity.
2. **Exploration claims:** The authors attribute enhanced exploration to BN’s stochasticity. While plausible, there is no explicit mechanism linking BN to exploration behavior; the effect could instead stem from smoother optimization or improved policy entropy. The claim should be more cautiously phrased.
3. **Reward vs Return terminology:** Some figures and text conflate “reward” with “final return.” It should be episodic return instead of reward.
4. **Algorithmic clarity:**
   - Algorithm 1 omits the replay-buffer storage step (`store(s_t, a_t, r_t, s_{t+1})`).
   - The clipping in Actor-I vs Actor-II differs without clear justification.
5. **Minor experimental details:**
   - In Figure 2, the replay buffer sizes labeled “5w/10w/100w” likely mean “5k/10k/100k.”
   - It would be helpful to explicitly state the default replay-buffer size used in main experiments (appears to be 100k).
6. **Causal interpretation:** Some causal language (e.g., “BN enhances exploration”) should be softened to correlation or indirect influence.


I think is a well-executed empirical study on a subtle but practically important issue in deep RL, so I'm leaning towards acceptance. My conrencs are addresabble and I'm willing to increase my score if the authors address them.

**Questions:**

- Why does Actor-I use different clipping parameters than Actor-II in Algorithm 1?
- In Figure 2, what is the buffer size used in the first row, and what is the batch size used in the second row?
- Why is the “EE” configuration excluded instead of shown as a degenerate baseline? The authors quickly mentioned that "the absence of BN statistics updates degenerates the BN layer into a fixed affine transformation" without detailed explanation.

---

> ### Author Response · Authors · 2025-11-20
> **Response to Reviewer #9xBN**
>
> We thank Reviewer 9xBN for recognizing the comprehensive study, clear motivation, important insights, practical relevance, and strong presentation of our paper. A detailed response is provided below.
>
> > w1: Terminology: The use of “training” and “evaluation” modes for BN may confuse readers, since both modes are used during RL training. Terms like *instantaneous BN* (using batch statistics) vs *running-average BN* might reduce ambiguity.
>
> Thank you for the valuable suggestions. In the revised manuscript we have uploaded, we added a clarification at the beginning of Section 3 regarding the distinctions between the training and evaluation modes, which helps readers better understand the use of different statistics during training.
>
> > w2: Exploration claims: The authors attribute enhanced exploration to BN’s stochasticity. While plausible, there is no explicit mechanism linking BN to exploration behavior; the effect could instead stem from smoother optimization or improved policy entropy. The claim should be more cautiously phrased.
>
> We appreciate your helpful suggestions and have accordingly updated the relevant discussions in Section 4.2 to express them in a more cautious manner.
>
> > w3: Reward vs Return terminology: Some figures and text conflate “reward” with “final return.” It should be episodic return instead of reward.
>
> We appreciate your helpful suggestions and have accordingly updated the relevant figures and the corresponding descriptions in the manuscript.
>
> > w4: Algorithmic clarity:
> > - Algorithm 1 omits the replay-buffer storage step (`store(s_t, a_t, r_t, s_{t+1})`).
> > - The clipping in Actor-I vs Actor-II differs without clear justification.
>
> We thank you for the valuable feedback. We have updated the description of Algorithm 1, focusing primarily on the update procedures. Regarding the clipping, please refer to our response to q1.
>
> > w5: Minor experimental details:
> > - In Figure 2, the replay buffer sizes labeled “5w/10w/100w” likely mean “5k/10k/100k.”
> > - It would be helpful to explicitly state the default replay-buffer size used in main experiments (appears to be 100k).
>
> We thank you for the valuable suggestion. We have updated the x-axis in Figure 2 to use a consistent unit of K for greater clarity and precision. Additionally, we have included detailed parameter descriptions in the Appendix I.
>
> > w6: Causal interpretation: Some causal language (e.g., “BN enhances exploration”) should be softened to correlation or indirect influence.
>
> We thank you for the valuable feedback. We have revised the relevant statements to present them in a more cautious manner.
>
> > q1: Why does Actor-I use different clipping parameters than Actor-II in Algorithm 1?
>
> Thank you for your question. We apologize for our oversight. In Algorithm~1, both Actor-I and Actor-II employ the same exploration noise $\epsilon \sim \text{clip}(\mathcal{N}(0, \sigma^2), -c, c)$. This design follows the standard procedure in TD3 [1], where noise is injected to encourage exploration, and clipping is applied to ensure that the perturbed target action remains close to the original policy output. Consequently, the update mechanism resembles that of Expected SARSA [2], in which the value estimate is learned off-policy while the injected noise in the target policy remains decoupled from the exploration policy. We again apologize for the mistake; this has been corrected in the revised manuscript, and we hope this clarification addresses your concerns.
>
> > q2: In Figure 2, what is the buffer size used in the first row, and what is the batch size used in the second row?
>
> Thank you for the question. For the experiments varying batch size, we fixed the replay buffer size to 100K for hopper_hop and quadruped_walk, and 10K for quadruped_run. For the experiments varying buffer size, the batch size was fixed at 256. Detailed descriptions of these parameters have been added to the Appendix I.
>
> > q3: Why is the “EE” configuration excluded instead of shown as a degenerate baseline? The authors quickly mentioned that "the absence of BN statistics updates degenerates the BN layer into a fixed affine transformation" without detailed explanation.
>
> Thank you for the question. BN updates its running statistics in training mode but not in eval mode. When using the EE setting, the running statistics are never updated and BN keeps normalizing with their initial values, effectively becoming a fixed affine transformation and thus failing to function as intended. Therefore, we exclude it. We have incorporated this explanation in the revised manuscript and hope this clarifies the issue.
>
> [1] Fujimoto S, Hoof H, Meger D. Addressing function approximation error in actor-critic methods[C]//International conference on machine learning. PMLR, 2018: 1587-1596.
>
> [2] Van Seijen H, Van Hasselt H, Whiteson S, et al. A theoretical and empirical analysis of expected sarsa[C]//2009 ieee symposium on adaptive dynamic programming and reinforcement learning. IEEE, 2009: 177-184.

---

### Meta-Review · Area_Chair_Ehya · 2025-12-21

**Summary:**

This paper investigagtes the usefulness of Batch Normalization (BN) in off-policy actor-critic algrithms It provides a set of experiments on the use of BN in such settings and analyzes failure modes as well as provides guidance on when and how to use BN in a deep reinforcement learning setting.

**Reviewer Concerns:**

There were several areas of concern. One, raised by all reviewers, was around clarifications, which the authors successfully provided.  Another one was about experiment choices and whether the experiments are sufficient to match the papers claim. The largest concerns were raised by reviewers 9xBN and Fd95 who made very similar and more general points about the lack of sufficiently strong motivation and direction of the paper and also questioned its significance.

**Reviewer Scores:**

While several concerns of the reviewers have been addressed, probably the main concern is about the relevance and significance of the work in the face of the CrossQ results. The authors are right to argue that their evaluation is covering a much broader scope by choosing DrQv2 as the basis of their analysis. This addresses one of the aspects of the reviewers concern. The other aspect was to question to what extent the usefulness of BN is actually surprising (Fd95). This concern seems to remain valid. Overall, I feel that the concerns of the two more critical reviewers are more foundational in nature and have not been resolved (although the authors did put in a good faith effort in adressing them).

---

### Decision · Program_Chairs · 2026-01-26

Reject